# Adaptation invariant concentration discrimination in an insect olfactory system

Doris Ling[1†], Lijun Zhang[2†], Debajit Saha[1†‡], Alex Bo-Yuan Chen[1§], Baranidharan Raman[1,2*]

[1]Department of Biomedical Engineering, Washington University in St. Louis, St Louis, United States; [2]Department of Electrical and Systems Engineering, Washington University in St. Louis, St Louis, United States

*For correspondence: barani@wustl.edu

[†]These authors contributed equally to this work

Present address: [‡]Department of Biomedical Engineering, Michigan State University, Lansing, United States; [§]Graduate Program in Neuroscience, Harvard Medical School, Boston, United States

Competing interest: The authors declare that no competing interests exist.

## eLife Assessment

This study addresses an **important** question in sensory neuroscience: how the olfactory system distinguishes decreases in stimulus intensity from decreases in neural responses due to adaptation. Based on a combination of electrophysiological and behavioral analyses, **solid** evidence establishes that neural coding changes differently between intensity reductions and adaptation, with intensity changes altering which neurons are activated while adaptation preserves the active ensemble but reduces response magnitude. Intriguingly, behavioral responses tend to increase as the neural responses decrease, suggesting that core features of the odor response persist through adaptation. While the experimental results are **convincing** overall, the conclusions will be strengthened by future work recording behavior and neural dynamics in the same animals.

**Abstract** Neural responses evoked by a stimulus reduce upon repetition. While this adaptation allows the sensory system to attend to novel cues, does information about the recurring stimulus, particularly its intensity, get compromised? We explored this issue in the locust olfactory system. We found that locusts' innate behavioral response to odorants varied with repetition and stimulus intensity. Counterintuitively, the stimulus-intensity dependent differences became significant only after adaptation had set in. Adaptation altered responses of individual neurons in the antennal lobe (neural network downstream to insect antenna). These response variations to repetitions of the same stimulus were unpredictable and inconsistent across intensities. Although both adaptation and intensity decrements resulted in an overall reduction in spiking activities across neurons, these changes could be disentangled, and information about stimulus intensity was robustly maintained by ensemble neural responses. In sum, these results show how information about odor intensity can be preserved in an adaptation-invariant manner.

## Introduction

The ability to adapt to sensory cues is key to the survival of many living organisms (*Castellucci et al., 1970*; *Clemens et al., 2018*; *Kadohisa and Wilson, 2006*; *Kaissling et al., 1987*; *McDiarmid et al., 2019*; *Parkinson and Revello, 1978*; *Pearson, 2000*; *Wu et al., 2022*). The goal of this sensory adaptation is to condense neural responses to a persisting or recurring stimulus in order to focus on novel cues. Such condensed neural responses are thought to increase system efficiency by reducing the metabolic requirements of information encoding (*Brenner et al., 2000*; *Laughlin, 1981*; *Smirnakis et al., 1997*; *Weber et al., 2019*). However, is sensory information lost when neural responses

are condensed during adaptation? While this computational task appears relatively elementary, any solution should satisfy two important requirements. First, attenuation of stimulus-evoked responses due to adaptation should not alter the identity of the stimulus. Second, information regarding the intensity of the stimulus may have to be preserved when odor intensity conveys information of behavioral importance. The latter requirement is paramount when sensory stimuli have different behavioral significance at varying intensities. For example, some odorants are attractive at lower intensities but switch to being aversive beyond certain threshold values (*Badel et al., 2016*; *Yoshida et al., 2012*; *Rong et al., 2017*). Sensitivity to intensity gradients has to be maintained for sustained periods to allow for navigation towards or away from the stimulus source (*Baker et al., 2018*; *Álvarez-Salvado et al., 2018*; *Gaudry et al., 2012*). Such biological contexts necessitate the preservation of odor intensity information across repeated encounters with the odorant. However, the mechanism by which information about stimulus intensity is robustly maintained is not fully understood.

Sensory adaptation occurs over both long- and short-time scales (*Weber et al., 2019*; *Whitmire and Stanley, 2016*; *Wilson, 2013*; *Bao and Engel, 2012*). Long-term adaptation requires gene transcription and protein translation that occurs on the time scale of hours to days (*Parkinson and Revello, 1978*; *Colbert and Bargmann, 1995*; *Kurahashi and Menini, 1997*; *Spehr et al., 2009*), whereas short-term adaptation occurs on a millisecond to minute time scale (*Clemens et al., 2018*; *Smirnakis et al., 1997*; *Martelli and Fiala, 2019*). Short-term adaptation results from a wide range of mechanisms with prominent models in the olfactory system being vesicle depletion and facilitation of lateral inhibition (*Martelli and Fiala, 2019*; *Diamond and Jahr, 1995*; *Murphy et al., 2004*; *Varela et al., 1997*; *Zucker, 1972*; *Stopfer and Laurent, 1999*; *Jafari and Alenius, 2021*; *Zufall and Leinders-Zufall, 2000*). While both vesicle depletion and lateral inhibition are likely to occur in the olfactory system, these complementary mechanisms will manifest in two different, but testable, outcomes. Vesicle depletion will impact the most prominent neurons with elevated odor-evoked responses, whereas facilitation of lateral inhibition would progressively suppress activity in the relatively weak responders. Irrespective of which mechanisms underlie olfactory adaptation, the neural responses evoked by a stimulus will change over time and across repeated encounters. This raises the question: Does adaptation then corrupt or alter information regarding the odorant?

The ambiguity that neural adaptation introduces has been documented in visual (*Fairhall et al., 2001*) and auditory systems (*Hildebrandt et al., 2015*; *Benda, 2021*). Prior studies have shown that odor identity is robustly maintained by the spatiotemporally patterned neural responses and can allow precise recognition across repeated encounters with the odorant (*Stopfer and Laurent, 1999*; *Stopfer et al., 2003*; *Saha et al., 2013b*; *Brown et al., 2005*). But is intensity information corrupted by the changes in neural activity due to sensory adaptation? Both reductions in stimulus intensity and sensory adaptation lead to a reduction in the overall spiking responses in most neural circuits. Can these neural response variations due to neural adaptation and intensity decrements be disentangled? We sought to examine this in the relatively simple insect olfactory system.

Our results show that odor intensity changes result in subtle variations in the overall combination of activated neurons. While sensory adaptation results in an overall reduction in the population neural responses, the activated ensemble is robustly maintained to allow simultaneous and precise decoding of odor identity and intensity. Using an innate behavioral assay, we show that physiological adaptation coincided with enhanced behavioral differentiation between odorant intensities. Thus, our work reveals a simple but elegant approach to achieving adaptation-invariant intensity coding and recognition in the insect olfactory system.

## Results

### Behavioral responses to an odorant vary with repetition and intensity

We began by examining how behavioral responses change with repetitions of an odorant. We used an innate behavioral assay where the presentation of certain odorants triggered an appetitive palp-opening response (POR) in starved locusts (*Figure 1a*; *Figure 1—figure supplement 1*). Typically, these palps are used to grab food such as a blade of grass and can also be triggered when locusts encounter certain food-related odorants. Hence, it has been used as a measure of the appetitiveness of an odorant (*Saha et al., 2017*; *Nizampatnam et al., 2022*; *Chandak and Raman, 2021*). For this study, we regarded the probability of an odor eliciting a POR (p(POR)) as an indicator for stimulus

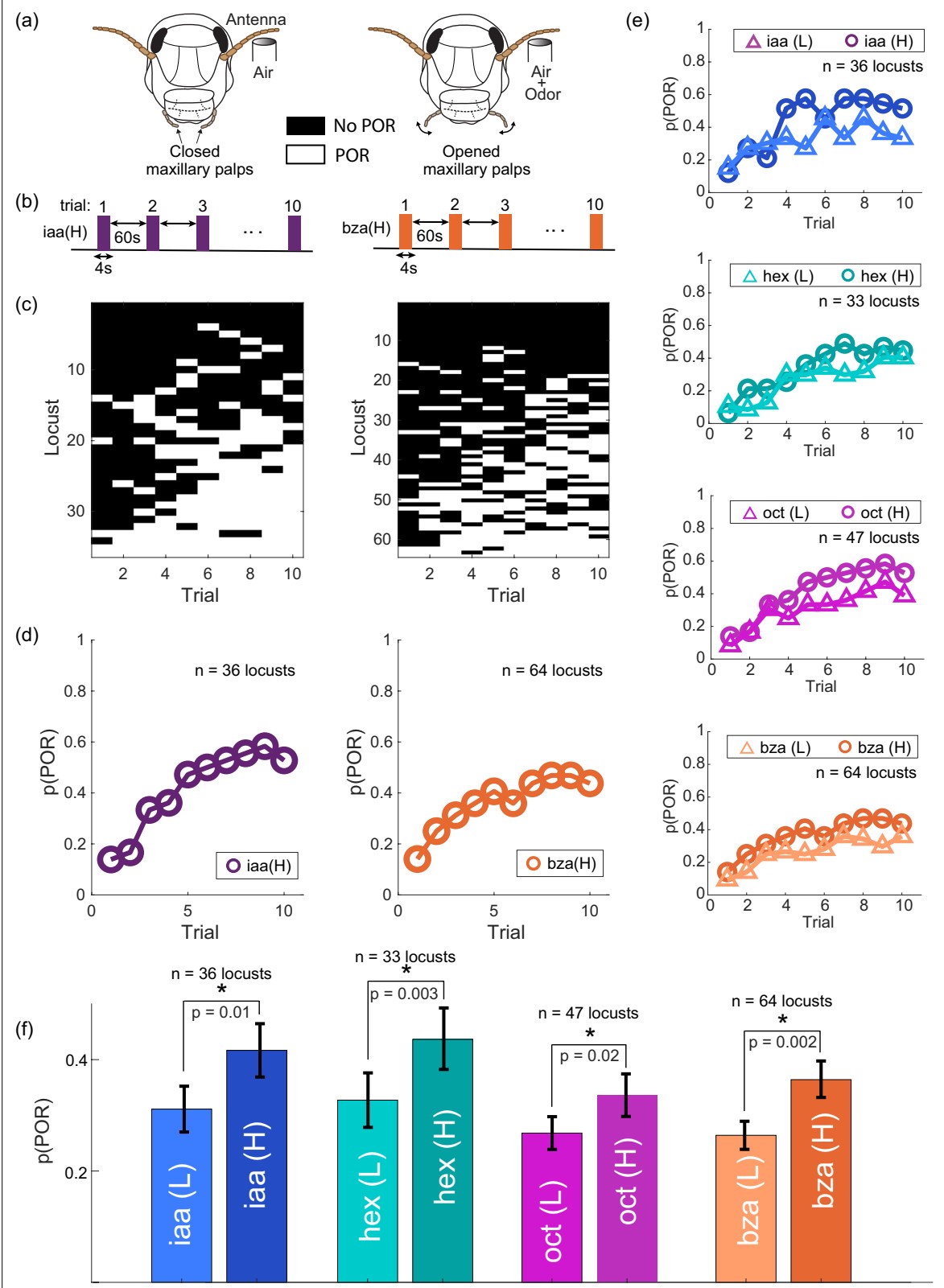

**Figure 1.** Behavioral response facilitation upon odor repetition. (**a**) Schematic of the innate palp-opening response (POR) experimental paradigm. Opening of the locust's maxillary palps within 15 s of the odor onset was considered a POR. (**b**) Schematic of the block odor stimulation protocol. Each block consisted of ten trials, and a 4 s odor pulse was presented in each trial. A 15 min no-stimulation period separated the blocks. (**c**) Response matrices are shown summarizing individual locust PORs (rows) for blocks of ten trials (columns). White boxes indicate PORs in a specific trial, while

*Figure 1 continued on next page*

*Figure 1 continued*

black boxes indicate the absence of PORs in that trial. Locusts were sorted such that the least responsive locusts are shown at the top and the most responsive ones are near the bottom. PORs varied between locusts across trials. The left matrix depicts responses of 36 locusts to isoamyl acetate. The right matrix shows the responses of 64 locusts to benzaldehyde. (**d**) The probability of PORs across locusts (p(POR)) is shown as a function of trial number for two odorants: iaa and bza at high intensities (1% v/v). (**e**) p(POR)s as a function of trial number is shown for four odorants (oct, hex, iaa, bza) at two different intensities (0.1% v/v – low and 1% v/v – high). The p(POR) of both odorant intensities increased over trials. Further, note that the p(POR) values for high-intensity odor exposures were notably higher than p(PORs) for low-intensity odor presentations. For this analysis, we computed the p(POR) for each locust at low and high concentrations of an odorant. This resulted in 36 paired values for isoamyl acetate, 33 paired values for hexanol, 47 paired values for octanol, and 64 paired values for benzaldehyde. We used a left-tailed t-test to check whether the p(POR) across all ten trials of lower intensity odor exposures was significantly smaller than the p(POR) during higher intensity presentations of the same odorant. (**f**) Bar plots compare the average P(POR) for the low- and high-intensity presentations of the same odor. Error bars indicate the standard error of the mean. p(POR) values observed at high and low intensities of an odorant were significantly different for all four odorants tested (p<0.05). n=36 locusts for iaa, n=33 locusts for hex, n=47 locusts for oct, and n=64 for bza.

The online version of this article includes the following video and figure supplement(s) for figure 1:

**Figure supplement 1.** POR responses of locusts across trials and odor intensities.

**Figure supplement 2.** POR responses of locusts to hexanol vs. paraffin oil control.

**Figure supplement 3.** Palp-opening responses in early vs. late trials.

**Figure 1—video 1.** A representative video showing locust palp-opening response.

https://elifesciences.org/articles/89330/figures#fig1video1

perception. The baseline p(POR), in the absence of any olfactory stimuli, was negligible. Further, presentations of puffs of paraffin oil (control) evoked only weak PORs (**Figure 1—figure supplement 2**). Note that the p(POR) was computed by averaging the behavioral responses across locusts for each trial or repetition number (**Figure 1b–d**). Intriguingly, we found that the p(POR) systematically increased with repetitions of the same odorant and stabilized after about five stimulus presentations.

Previous results have shown that the POR is a function of both stimulus identity and intensity (**Saha et al., 2017**; **Nizampatnam et al., 2022**; **Chandak and Raman, 2021**). Therefore, we presented each locust with randomized blocks of the same odorant at either high (H, 1% v/v) or low (L, 0.1% v/v) intensities (**Figure 1e**). Our results indicate that the p(PORs) evoked by high and low stimulus intensities of an odorant were significantly different for all four odorants in our panel. (**Figure 1f**; p≤0.05; one-sided t-test). Notably, the behavioral responses to high and low concentration exposures of the same odorant were not significantly different in the earlier trials (**Figure 1—figure supplement 3**; p>0.05 for hex, iaa, and oct; one-sided t-test). However, the p(POR) became significantly different during the later trials (p≤0.05 for all four odorants; one-sided t-test; trials 6 through 10). In other words, high and low stimulus intensities were more behaviorally discriminable in the later trials. Hence, these results counterintuitively suggest that behavioral responses to an odorant at two different intensities were different after multiple encounters with the same stimulus.

## Odor-evoked response adaptation leads to complex changes in individual neurons

How do neural responses change as a function of stimulus repetition? To understand this, we examined the odor-evoked responses of individual projection neurons (PNs) in the locust antennal lobe as different odorants at two different intensities were repeatedly presented (**Figure 2a**). Odor-evoked responses in individual PNs varied subtly across trials (**Figure 2b**). In general, most PN responses to the odorant were stronger in the first trial and reduced during later trials of the same stimulus. Notably, the changes in PN responses in two different blocks of trials, where the same odorant was presented, showed repeatable changes in odor-evoked spiking patterns and the overall response strength (**Figure 2c**).

Next, we examined how individual PNs adapted during stimulus repetitions. Based on existing literature, two widely considered models of adaptation in the olfactory system are vesicular depletion and facilitation of lateral inhibition. While the vesicular depletion model predicts that the strongest firing neurons would adapt the most, in the lateral inhibition facilitation model, the weakly activated neurons would progressively be suppressed more (**Figure 2d**). To test these two opposing but complementary theories, we calculated the total change in the total spike counts observed during the 25th and first odor presentations (**Figure 2e**, *y-axis; positive values indicate stronger response in the*

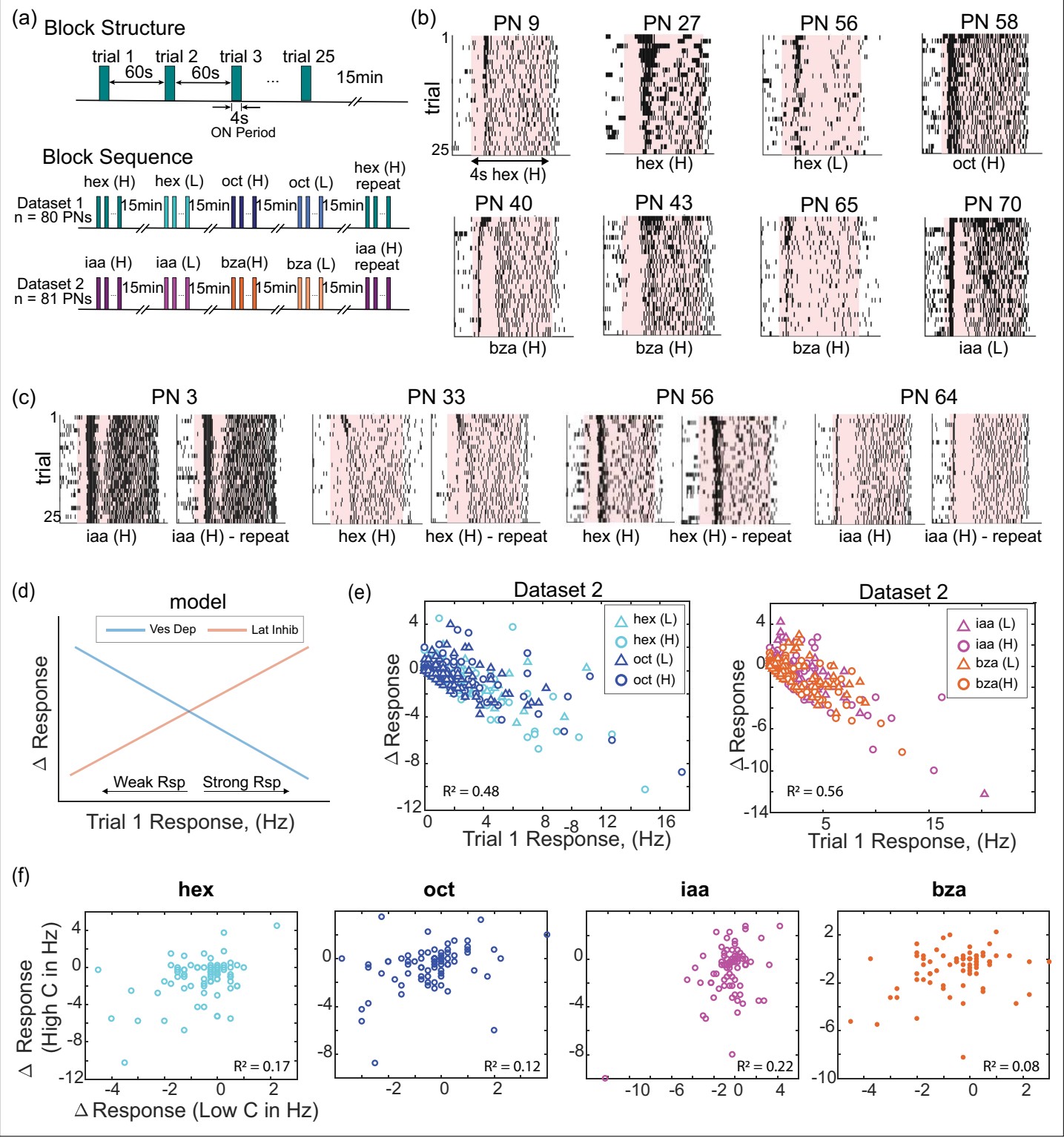

**Figure 2.** Ensemble neural activity systematically changes across repeated encounters with an odorant. (**a**) Schematic of the olfactory stimulation protocol. Each block consisted of 25 trials with a 4-s odor pulse delivered in each trial. The inter-trial interval was 60 s. Two datasets were collected. Each dataset consisted of five randomized blocks of four odorants (dataset 1: hex (H), hex (L), oct (H), oct (L), hex (H)-repeat; dataset 2: iaa (H), iaa (L), bza (H), bza (L), iaa (H)-repeat). A 15-min no-odor stimulation period separated blocks of trials. (**b**) Raster plots of eight representative projection neurons (PNs) in the locust antennal lobe are shown. Spiking activities are shown for 25 trials (rows) with earlier trials shown at the top and later trials at the bottom. The shaded region indicates the 4-s odor stimulation period, and the identity of the stimulus is indicated in each plot. (**c**) Raster plots

*Figure 2 continued*

are shown for four representative PNs during two blocks of trials. The same odorant was presented in both blocks. Note that spiking activity changes are repeatable across the block of trials. (**d**) Schematic showing how vesicular depletion and lateral inhibition facilitation models would change spiking activity in individual neurons. The y-axis represents the change in response over trials (25th trial response – 1st trial response). Positive numbers indicate that the last trial had a stronger response, hence the response increase. Negative numbers indicate that the first trial had a stronger response, hence response reduction. Along the x-axis, the response in the first trial is shown. Vesicular depletion should impact the strongly activated neurons more, whereas lateral inhibition facilitation should progressively suppress weak responders. (**e**) Following the schematic in panel **d**, response changes observed in PNs are shown. All PNs were included in this plot. Different colors and symbols are used to denote the identity of the odorant and its intensity (low-intensity trials – triangles and high-intensity trials – circles). As can be noted, the $R^2$ values are ~0.48–0.56 indicating that the vesicle depletion facilitation model captures the adaptation trends in our datasets to a certain extent. (**f**) Changes in PN spiking activity over trials during high-intensity odor exposures (y-axis) are plotted against response changes observed for the same PN during low-intensity exposures of the same odorant. The poor correlation values (hex, $R^2$=0.17; oct, $R^2$=0.12; iaa $R^2$=0.21; bza $R^2$=0.08) indicate that reductions in neural response amplitude for one odor intensity do not model reductions in response amplitude for another odor intensity.

later trial and hence response facilitation, whereas negative values indicate reduction of odor-evoked responses). We plotted these response changes against the spike response observed in the first trial (*Figure 2e*, *x-axis*). If vesicular depletion was the predominant form of adaptation, there would be a negative correlation with the response strength in the first trial. Whereas a positive correlation would indicate that lateral inhibition was the predominant form of adaptation. As can be noted, across both datasets with two concentrations of the four odorants tested, the vesicle depletion hypothesis appeared to capture some of the adaptation dynamics in our datasets ($R^2$=0.48, 0.56, respectively).

Are trial-to-trial changes in stimulus-evoked neural responses at two different concentrations of the same odorant correlated? If this is the case, then it would be reasonable to expect neurons that adapt more at the higher concentration to also change in a similar fashion at a lower concentration and *vice versa*. To assess this, we calculated and plotted the change in spike counts for each PN at both high and low concentrations of the same odorant (*Figure 2f*). Notably, the PNs with the greatest response change during repetitions of the higher-intensity odor exposures were not the ones that adapted heavily when encountering the same stimulus at a lower intensity ($R^2$=0.17 (hex), 0.12 (oct), 0.21 (iaa), 0.08 (bza)). In sum, these results indicate that change due to adaptation in individual neurons is not a simple function of their response strength. Further, how individual neurons adapt is a function of both odor identity and intensity.

## Adaptation and intensity decrements both reduce ensemble neural response strength

What is the overall change in the spiking response across the neural ensemble? To examine this, we calculated the average spiking response across all PNs recorded and plotted the spike counts as a function of time for each trial (*Figure 3a*). Expectedly, the overall response strength was stronger in the first trial and reduced when the same stimulus was repeatedly encountered. A similar reduction in overall response strengths was also observed when the odorant intensity was reduced (*Figure 3b*).

Are the response reductions during stimulus repetition comparable to those observed with stimulus intensity decrements? To determine this, we plotted the total spike counts summed across all PNs over the entire stimulus presentation duration and plotted it as a function of the trial number (*Figure 3c*). We compared the changes over trials with the total spike counts observed during the high (H) and low (L) intensities of the same odorant. Notably, the total PN ensemble spike counts decreased due to both stimulus intensity reduction and repetition of the stimulus. Do these results indicate that adaptation could potentially confound information regarding stimulus intensity?

## Adaptation-invariant encoding of odor intensity

Could the information regarding stimulus intensity be robustly encoded in the population neural responses? As noted earlier, both spiking activities in individual neurons and the volume of spikes at the ensemble level vary across trials. If adaptation alters the overall spiking activity amplitude (i.e., vector length; *Figure 4a*), while changes in stimulus intensity alter the combination of activated neurons (i.e. vector direction; *Figure 4b*), then information about odor identity and intensity can be preserved in an adaptation-invariant fashion. To test whether the recorded ensemble of neurons in our datasets encoded information in such an adaptation-invariant fashion, we first visualized the high-dimensional

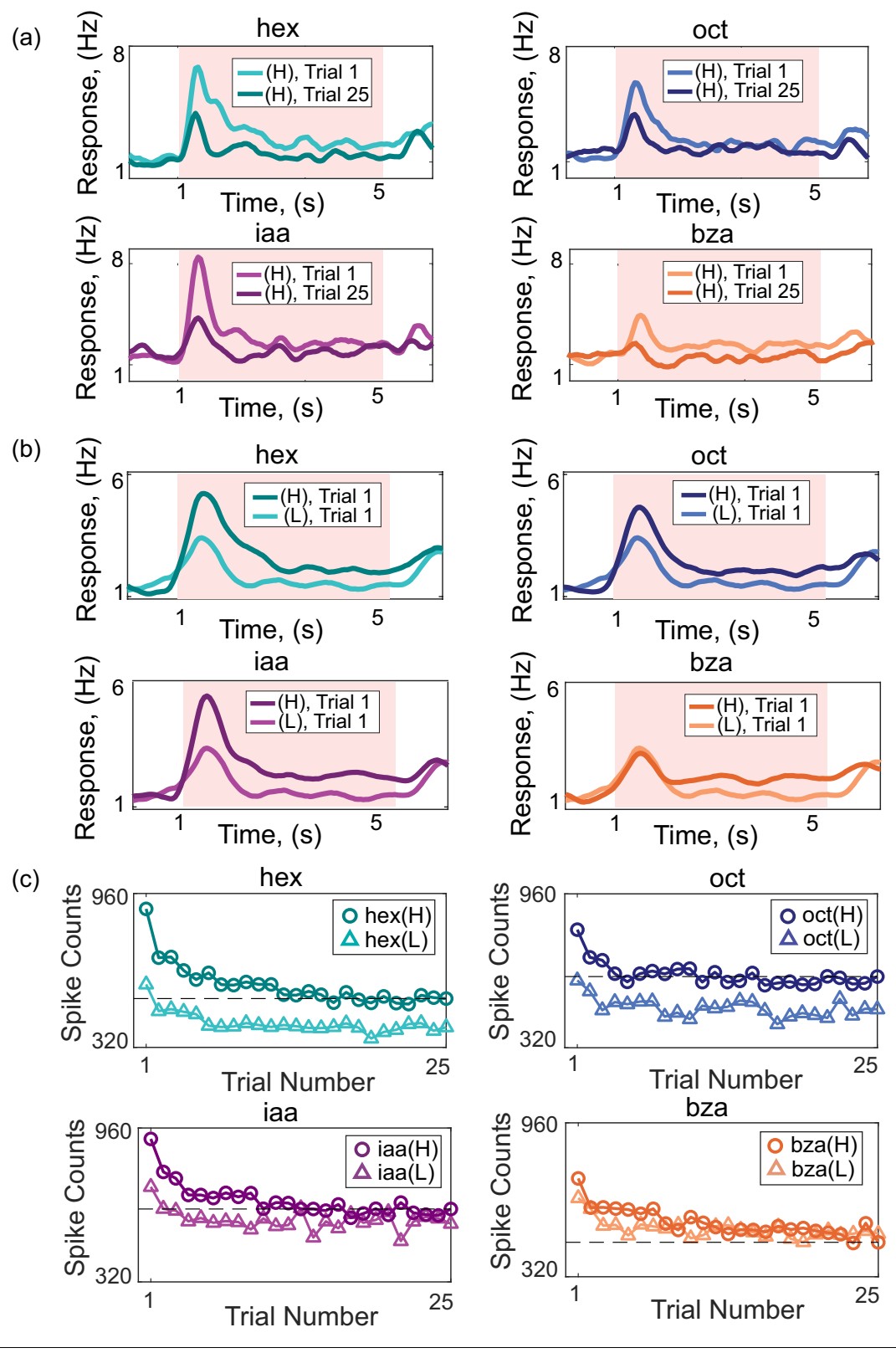

**Figure 3.** Stimulus repetition and intensity decrements reduce spiking responses. (**a**) Peristimulus time histograms (PSTHs) across all PNs (hex and oct, n=80 PNs; iaa and bza, n=81 PNs) for each odorant. PSTHs of trials 1 and 25 are shown. (**b**) PSTHs across all PNs are compared between high and low-intensity odor exposures. Response during the first trial is shown. (**c**) Summed spike counts (across PNs and four-second odor presentation) are

*Figure 3 continued on next page*

*Figure 3 continued*

calculated and shown as a function of the trial number. The dotted line indicates the spike count of the 25th trial of the high-intensity odor exposure.

neural activities in each trial using a dimensionality reduction approach (see Methods). We plotted the ensemble responses in each 50ms time bin during the odor presentation time window (4 s) and linked them based on the order of their occurrence to generate trial-by-trial odor response trajectories (*Figure 4c and d*). Note that each trial generated a single loop response trajectory after dimensionality reduction. Further, neural responses in different trials evoked closed-loop trajectories that systematically reduced in length with increasing repetition but maintained their direction, indicating that the combination of neurons activated was mostly conserved. Different odorants evoked responses that were markedly different in the combination of activated neurons, and therefore, their response trajectories were oriented in different directions. Notably, different concentrations of the same odorant also

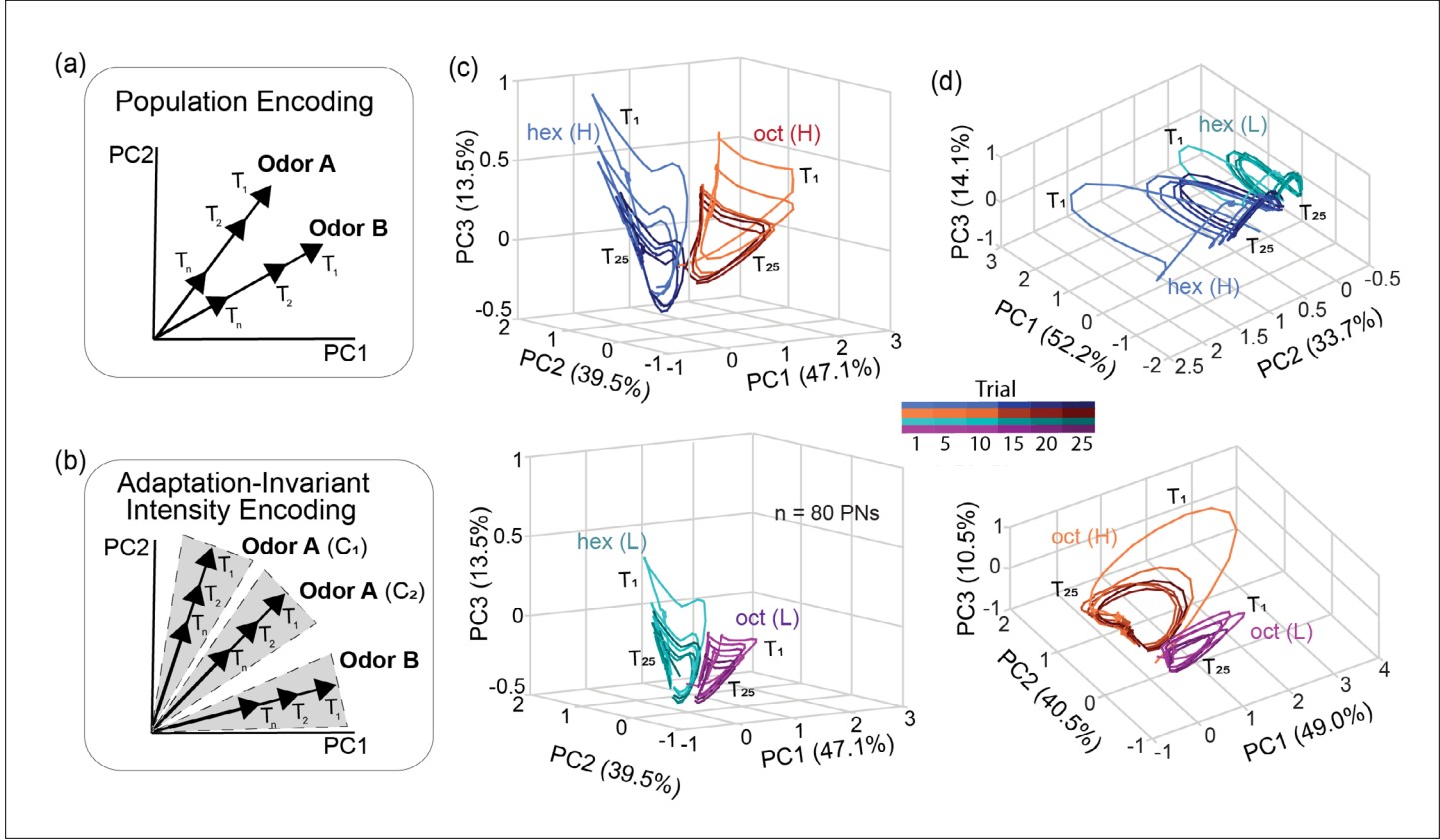

**Figure 4.** Odor identity and intensity information are maintained across trials. (**a**) A schematic showing how the ensemble neural activity might change between two odorants and across multiple trials or repetitions. The combination of neural activation should differ between different odorants, and therefore, the odor identity should be represented by population response vectors that differ in their direction. Repetitions should reduce response strength without altering the combination of neurons activated. If this were the case, the later repetitions of the same odorant would evoke a response that can be represented by vectors that maintain directions while progressively becoming shorter in length or magnitude. (**b**) A schematic coding scheme for achieving adaptation-invariant intensity coding. The combination of neurons activated changes markedly with odor identity and subtly with odor intensity. These responses become less intense without altering the combination of neurons activated. If this were the case, then the ensemble vector direction would change with both odor identity and intensity, and the vector direction would be robustly maintained even though the vector length continues to change with repetition. (**c**) Trial-by-trial odor-evoked ensemble PN response trajectories after dimensionality reduction are shown for hex and oct. Only high-concentration exposures of both odorants are included in this plot. For each odor, the temporal response trajectories for the 1st, 5th, 10th, 15th, 20th, and 25th trials are shown (color gradient from light to dark as trial number increases). Note that the odor-response trajectories become increasingly smaller during later trials, but the direction of the trajectories is robustly maintained. (**d**) Trial-by-trial response trajectories for low-concentration exposures of the hex and oct are shown. Similar convention as in panel **c**. (**e**) Similar plot as in panel **c**, but now comparing the odor-evoked response trajectories elicited by hex at high and low intensities. (**f**) Similar plot as in panel **d**, but comparing the response trajectories elicited by oct at high and low intensities.

generated response trajectories that subtly varied in direction. As a direct consequence, even though the overall spiking activities reduced over trials, the odor response trajectories reliably maintained their direction in all the trials. Hence, information about odor identity and intensity was robustly maintained across all trials.

To quantify these observations, we performed a correlation analysis using high-dimensional PN response vectors. If a similar combination of PNs (high-dimensional activity vectors) were activated in two different trials, then their correlation would be high. Correlations across different trials and concentrations of the same odorant are shown in *Figure 5a*. Confirming results from the dimensionality reduction analysis (*Figure 4*), we found odor-evoked responses across different trials of an odorant at a particular intensity were highly correlated (diagonal blocks on each panel). The similarity between odor-evoked responses evoked by different concentrations of the same odorant was considerably less correlated (off-diagonal blocks). A hierarchical clustering analysis revealed that the odor-evoked

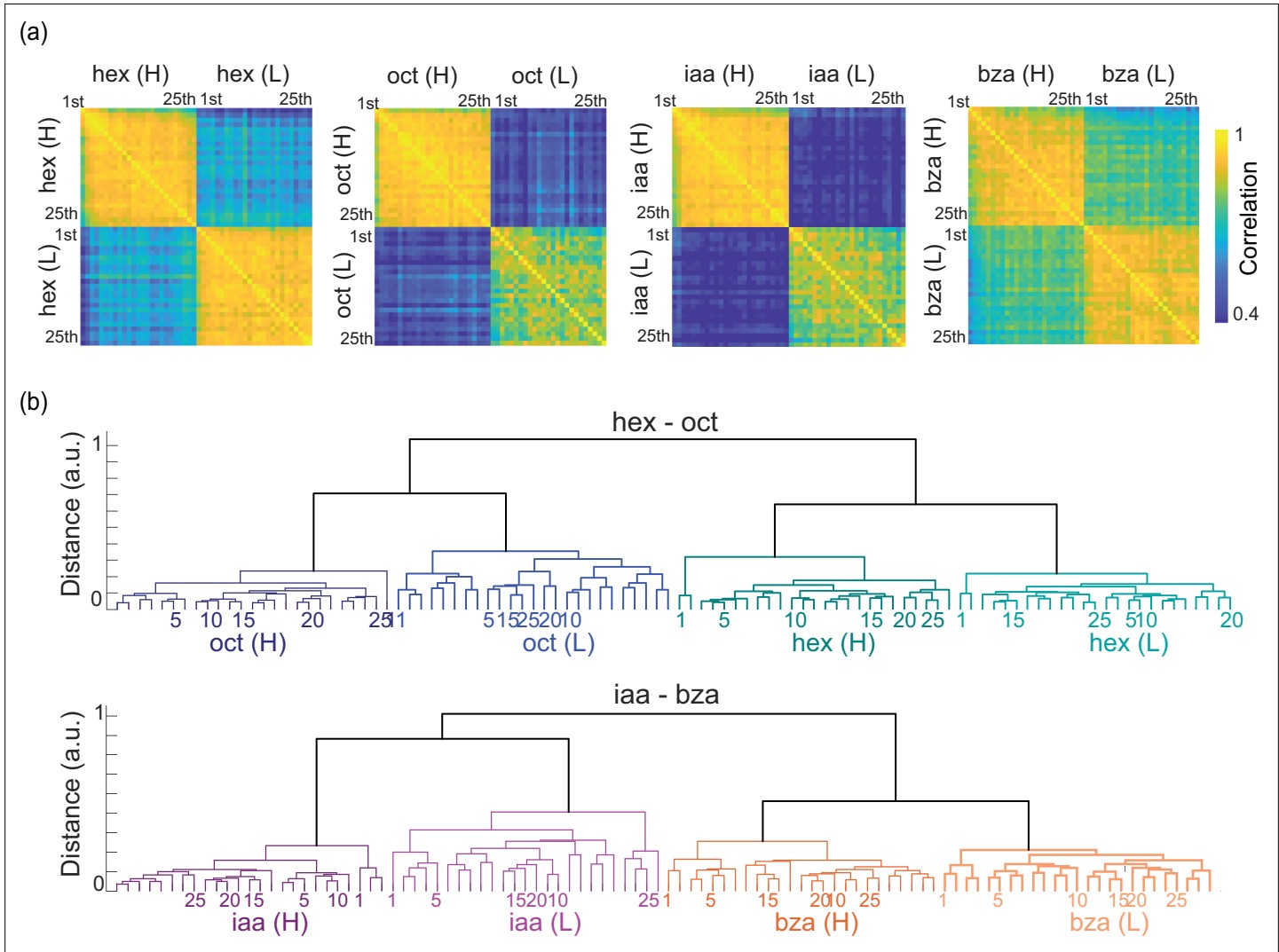

**Figure 5.** High and low stimulus intensities activate distinct ensembles. (**a**) Correlations between neural responses observed in different trials are shown. Each pixel/matrix element represents a similarity between mean neural responses in one trial versus those in another trial. Diagonal blocks reveal the correlation between trials when the same odorant at a specific intensity was repeatedly presented. (**b**) A dendrogram was generated using a correlation distance metric comparing trial-by-trial ensemble spiking activities evoked by two different stimuli at two different intensities (see Methods). Two major response clusters that correspond to stimulus identity and intensity were identified. The number at the leaf node represents the trial number.

The online version of this article includes the following figure supplement(s) for figure 5:

**Figure supplement 1.** Classification analysis confirms robust adaptation-invariant odor recognition using ensemble neural activity patterns.

**Figure supplement 2.** Combinatorial response patterns across two intensities of the same odorant.

responses were nicely clustered based on odor identity and then by intensity (*Figure 5b*). These observations were further corroborated by a classification analysis (*Figure 5—figure supplement 1*).

In sum, our results reveal that a combinatorial code could encode information regarding odor identity and intensity in an adaptation-invariant fashion. Since the same information about a stimulus is represented with fewer spikes in the later trials, we conclude that adaptation refines the odor codes by making them more efficient.

## Neural response suppression vs. behavioral output facilitation

Neural responses evoked by an odorant reduced with repetition (*Figure 3*). However, as we noted earlier, the behavioral responses increased upon stimulus repetition (*Figure 1*). This negative correlation is evident when the neural responses (total spiking activities evoked across neurons) were plotted against the behavioral POR response probability during a particular trial (*Figure 6*). This negative correlation between neural suppression and behavioral enhancement was observed for all odorants and both concentrations examined. It is worth noting that the last trial of the high-intensity exposure evoked neural responses that were comparable in strength to the first trial of the low-intensity exposure of the same stimulus. Yet, the behavioral responses observed during these two trials were markedly different. Hence, our results also indicate that the probability of the POR response is not simply a function of the total number of spikes elicited.

## Neural and behavioral changes due to adaptation are odor-specific

Finally, we wondered if the neural response reduction and behavioral enhancements were a global, non-specific state change in the olfactory system brought about by the repetition of any odorant, or are the observed neural and behavioral response changes odor-specific. To examine this, we used a 'catch-trial paradigm'. (*Figure 7a*) In these sets of experiments, we repeatedly presented an odorant to induce adaptation. After a substantial reduction in the neural response or stabilization of the behavioral response, a deviant or catch stimulus was presented. Persistence of the observed reduction in neural responses and enhancement in behavioral responses when the catch stimulus is introduced would reveal that the adaptation-induced changes are not odor-specific.

We used the catch-trial experiments with two different catch stimuli. Our neural recordings reveal that when iaa (ester) was used as the catch stimulus after repeated presentations of hex (alcohol), the neural response strength recovered and increased to an overall higher spike count (*Figure 7b*). The observed catch-iaa response strength matched the unadapted response level to iaa when it was separately presented following a 15 min reset window (i.e. *Figure 7b*, trial# 31). In contrast, when apple (a complex mixture) was used as the catch stimulus, the neural responses did not recover to higher levels indicating significant cross-adaptation (*Figure 7b*). Nevertheless, in both cases, the ensemble responses evoked by the catch stimulus were highly similar to the unadapted responses they usually generate (*Figure 7c*). Therefore, these results indicate that there is stimulus specificity to the adaptation-induced changes in neural responses and that cross-adaptation does not corrupt the identity of the deviant stimulus.

Behaviorally, we chose odor pairs where the component odorants had markedly different levels of POR responses (*Figure 7d*). Repeated presentations of an odorant (bza or iaa) increased the p(POR) across the locusts as observed before (*Figure 7e*). Notably, when the catch stimulus was presented, the behavioral responses dropped close to the first-trial p(POR) elicited by that odorant (*Figure 7f*; dotted line). These findings were also observed when the recurring and catch odorants were switched (*Figure 7f*).

In sum, these neural and behavioral results indicate that the observed neural and behavioral response changes brought about by the repeated encounters with an odorant are at least partially odor-specific and are not global, non-specific changes in the olfactory system.

## Discussion

Sensory systems often have to deal with constraints that are of an opposing nature. For example, the ability to maintain stable representations is key to robust recognition of sensory stimuli (*Stopfer et al., 2003*; *Storace and Cohen, 2017*; *Asahina et al., 2009*). Often, variations in the external environment, such as humidity and temperature, may introduce changes in neural responses that have to be filtered

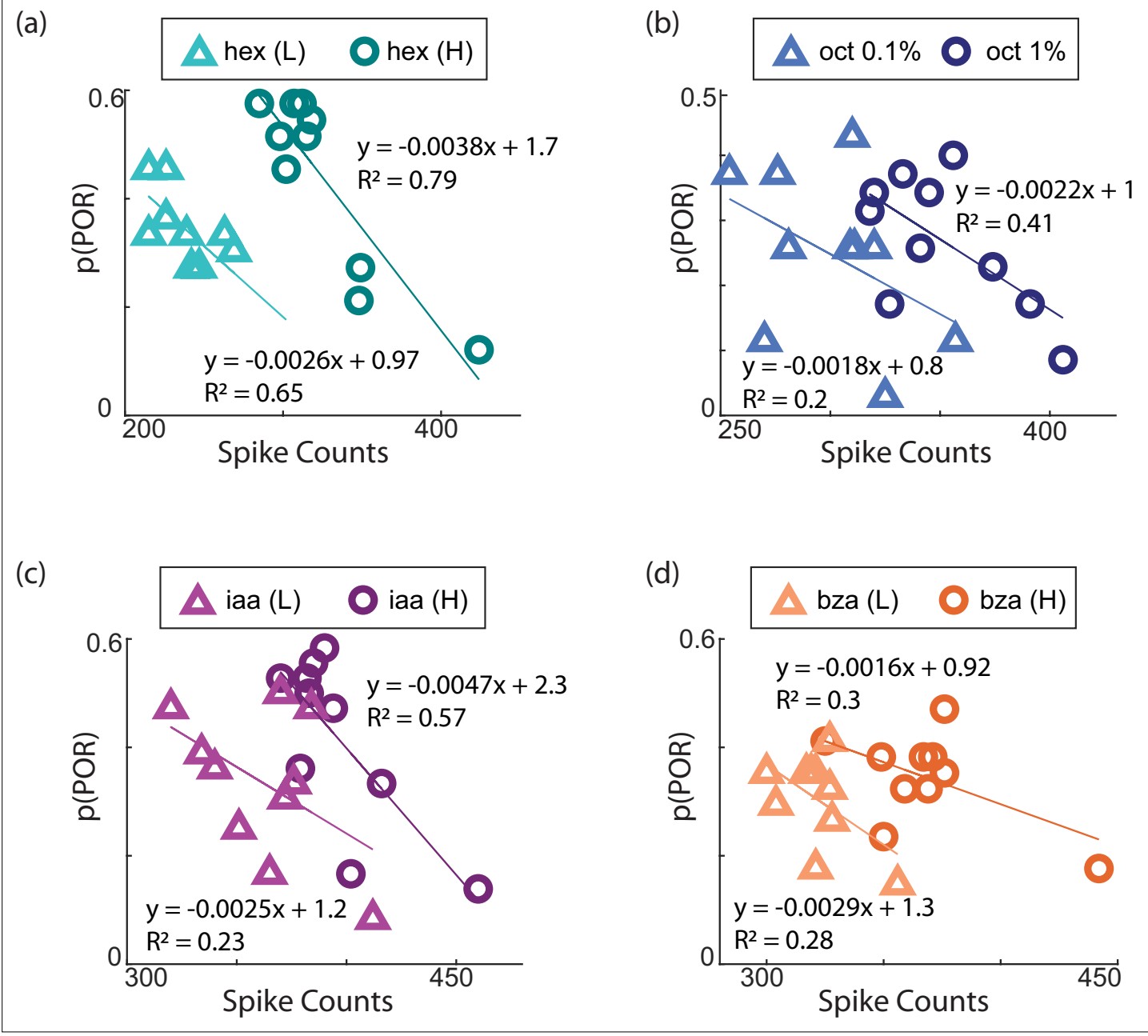

**Figure 6.** Neural adaptation inversely correlates with behavioral facilitation. (**a**) The probability of odor-evoked POR (P(POR)) for a given trial is plotted against the total spike counts elicited during the trial or repetition number. Therefore, each point represents a single trial. Symbols with light and dark colors are used for differential trials of high and low-concentration odor exposures. The line represents the regression fit between the behavioral and neural responses (hex (L), $R^2$=0.65; hex (H), $R^2$=0.79). The negative slope of the regression line indicates that while the neural responses diminish over trials, the behavioral responses increase over trials. (**b–d**) Similar plots as in panel **a** but for oct (L/H), iaa (L/H), and bza (L/H).

out to preserve the identity of the stimuli. Additionally, the strength or intensity of the stimulus can also change widely. Compressing or even removing these variations might be necessary to recognize an odorant independent of its intensity (*Stopfer et al., 2003*; *Storace and Cohen, 2017*; *Asahina et al., 2009*; *Uchida and Mainen, 2008*; *Wilson et al., 2017*; *Bolding and Franks, 2018*). Alternately, in certain cases, it might also become important to maintain some of these variations so they can be exploited for guiding or altering the behavioral response. As an example, it might be worth noting that many chemicals that are pleasant at lower intensities can become repelling or even harmful to the organism at extremely high intensities (*Badel et al., 2016*; *Yoshida et al., 2012*; *Rong et al., 2017*).

Hence, the behavioral responses should accordingly vary with intensity. Further, many organisms have shown that they can follow an odorant to its source (source localization). Such behaviors are of ecological importance as they allow the organisms to find the food source or potential mates. In all these cases, it becomes important to maintain variations with stimulus intensity to ensure the organisms are able to approach the odor source. In our earlier studies, we have focused on examining how robust odor recognition is achieved (*Saha et al., 2013b*; *Nizampatnam et al., 2022*; *Nizampatnam et al., 2018*). Here, we examined the latter problem of how to maintain information about the stimulus intensity while the responses in the neural circuits are becoming weaker due to adaptation.

Consistent with earlier studies (*Stopfer and Laurent, 1999*; *Twick et al., 2014*; *Stopfer et al., 1997*), our data revealed that the response of individual PNs in the antennal lobe varied with the repetition of the same stimulus. While a reduction in odor-evoked responses was observed in many neurons, an increase in spiking activity and changes in temporal firing patterns were also observed. Notably, these changes were not random but highly repeatable (*Figure 2c*). Hence, these results indicate that the effect of adaptation in a neural circuit is highly consistent and reliable.

In addition to spike rate reduction, oscillatory synchronization, and inter-neuronal coherence build-up in the locust antennal lobe over repeated encounters with an odorant (*Stopfer and Laurent, 1999*; *Laurent and Davidowitz, 1994*; *MacLeod and Laurent, 1996*). This short-term memory was not observed in the antenna but endured in the antennal lobe for several minutes after the termination of the stimulus. Similar results were also reported in the mouse olfactory bulb (*Storace and Cohen, 2017*). The inputs from sensory neurons were shown to be consistent across repetition, whereas the outputs of the olfactory bulb were systematically reduced with stimulus repetition. These results suggest that loci of short-term adaptation, at least when the stimulus is discontinuous and recurring, are in the antennal lobe/olfactory bulb circuitry. Although the precise mechanism for achieving the same is not fully understood (*Twick et al., 2014*; *Ramaswami, 2014*).

What is the computational significance of this neural adaptation? Sensory adaptation has been implicated in several important computations such as high-pass filtering, matching neural response to stimulus statistics, generating sparser codes, and optimal representations (*Fairhall et al., 2001*; *Benda, 2021*; *Gorur-Shandilya et al., 2017*; *Rinberg et al., 2006a*; *Rinberg et al., 2006b*). In the olfactory system, adaptation has also been suggested to stabilize neural representation such that recognition of odor identity from ensemble neural responses improved after repeated encounters with the stimulus (*Stopfer and Laurent, 1999*). However, adaptation also introduces a potential confound whereby the absolute neural response strength becomes an unreliable indicator of stimulus strength. Our results extend these earlier studies by revealing how this potential confound could be resolved in the olfactory system.

Increases in stimulus intensity have also been shown to result in higher response amplitudes in firing neurons and the recruitment of additional activated neurons (*Benda, 2021*; *Stopfer et al., 2003*; *Asahina et al., 2009*; *Silbering and Galizia, 2007*). While our results show that this is indeed true, we also found that several neurons responded preferentially during the lower concentration exposures but did not respond when the higher intensity of the same stimulus was presented (*Figure 5—figure supplement 2*). Hence, the activated combination of neurons differed both with stimulus identity and intensity. Furthermore, even though adaptation reduced the overall spiking activities across neurons, the combination of neurons activated was robustly maintained. In other words, the population response vector direction (determined by the combination of neurons activated) was consistent. Only the vector length (determined by the total spikes across neurons) was reduced with the repetition of the stimulus. This simple approach was sufficient to encode the information about the odor intensity in an adaptation-invariant fashion. Additionally, since equivalent information about stimulus identity and intensity was preserved with fewer spikes, neural adaptation can be considered as a method of optimizing the neural representation (*Clemens et al., 2018*; *Kadohisa and Wilson, 2006*; *Brenner et al., 2000*; *Laughlin, 1981*; *Fairhall et al., 2001*; *Hildebrandt et al., 2015*).

Does the observed persistence of concentration information in the neural activities after adaptation translate to observable differences in behavioral outcomes? Using an innate POR, we examined how behavioral responses varied with stimulus repetition. Unlike the neural responses, we found that the PORs increased with repetition and, after about five trials, settled into odor-specific levels of responses. Notably, the PORs in the later trials were significantly higher for higher stimulus intensities for most of the odorants used. No such changes were observed during repeated presentation of

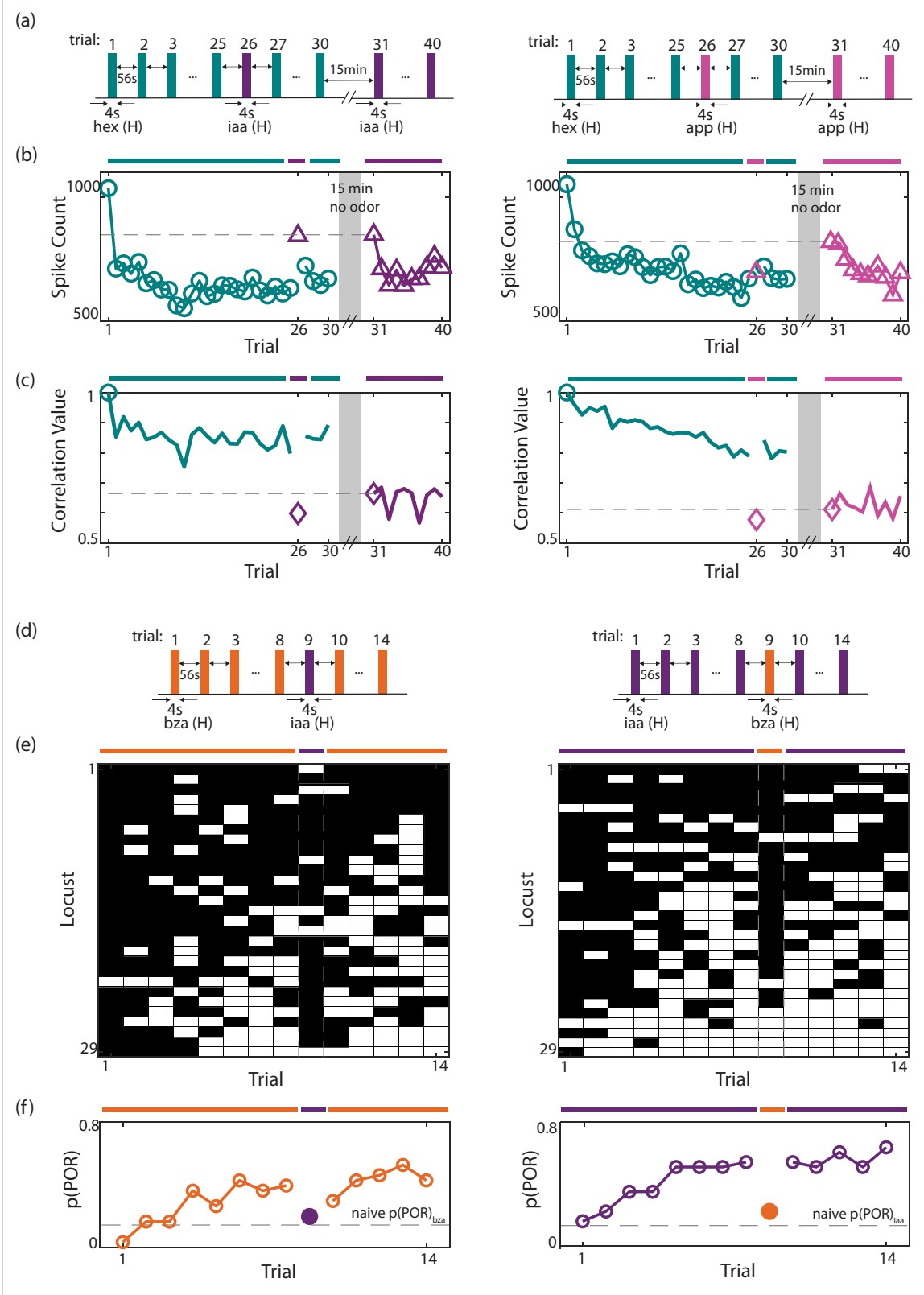

**Figure 7.** Adaptation results in odor-specific neural response reductions and behavioral output facilitation. (**a**) Two blocks of trials were used. First, a block of 30 trials where one odorant (*hex*) was presented in all trials except the 26th trial (the catch trial). During the catch trial, a deviant stimulus (*iaa* or *app*) was presented. After a 15 min no-odor reset period, a second block of 10 trials of the deviant stimulus was presented (trials 31–40). This was done to determine the unadapted (first trial) and adapted (later trials) responses of the same set of PNs to the stimulus used in the catch trial. (**b**) Summed

*Figure 7 continued on next page*

*Figure 7 continued*

spike counts (during 4 s odor presentation period) across all PNs were calculated and shown as a function of trials (*hex* repetitions). Repeated presentations of odor A resulted in a substantial reduction in odor-evoked spike counts. However, the presentation of the deviant odor (*iaa*) during the catch trial (trial # 26) resulted in a marked increase in spike counts. The response strength was similar to the non-adapted responses (Trial # 31) for *iaa* (H). (right) Similar results are shown when *hex* was repeatedly presented and the *app* was used as the deviant stimulus. Note that the response to the app during the catch trial did not recover as was noted for iaa (left). (**c**) Correlations between odor-evoked responses with hex response in the first trial are shown as a function of trial number. (**d**) A similar catch-trial paradigm was followed for behavioral experiments. Each locust was presented with a repeating odorant for eight trials. A deviant stimulus was presented in the ninth trial. This was followed by six more repetitions of the recurring stimulus. (**e**) The response matrix summarizing the POR responses of individual locust PORs is shown. Same convention as in *Figure 1*. (**f**) The P(POR) across locusts was calculated and plotted as a function of the trial number. Notably, when the deviant stimulus was presented, there was a marked decrease in P(POR). The dotted line indicates the unadapted P(POR) value for the deviant stimulus.

paraffin oil (the solvent used for diluting odorants; *Figure 1—figure supplement 2*). This suggests that the increases in behavioral responses with repetition were driven by the presence of the olfactory stimuli and not just the mechanical component (i.e. flow or pressure changes associated with stimulus presentations). Furthermore, our results showed that locusts are better able to behaviorally differentiate stimulus intensity after adaptation.

Alternately, adaptation due to the repetition of an odorant could alter the perceived intensity of the recurring odorant or other related test odorants. The compromise of stimulus intensity information due to adaptation has been observed in behavioral discrimination tasks in rats and even in psychophysical studies in humans (*Wojcik and Sirotin, 2014*; *Cain, 1970*). Such a result can arise when the combinatorial response patterns evoked by the adapting and the test stimulus have significant overlap. However, our results indicate that when changes in odor intensities alter the combination of neurons activated, it is feasible to maintain information about odor intensity levels in a robust fashion.

Finally, we note that the anticorrelated link between neural suppression and behavioral enhancements (called 'repetition priming') is well established in other animal models (*Grill-Spector et al., 2006*). How can weaker neural responses lead to enhanced behavioral outcomes? Our results indicate that the antennal lobe neural responses in later trials were not sharpened by pruning out weaker responses (*Figure 5—figure supplement 2*). The weakening of the response strength could potentially lead to a sparser response in the downstream mushroom body. In addition, as we noted earlier, neural response synchrony increases with stimulus repetition. Whether weaker but synchronized activity in neurons drives sparser and more selective responses, which then produce enhanced behavioral outcomes, remains to be carefully examined.

## Methods
### Odor stimulation

For delivering odorants, we followed a protocol described in our earlier work. Briefly, odorants were diluted in paraffin oil to either 1% or 0.1% concentration (v/v) and sealed in glass bottles (60 ml) with an air inlet and outlet. A pneumatic picopump (WPI Inc, PV-820) was used to displace a constant volume (0.1 L/min) of the static headspace above the diluted odor-mineral oil mixture into a desiccated carrier air stream (0.75 L/min) directed toward one of the locust's antennae. A vacuum funnel placed behind the locust preparation continuously removed the delivered odors.

The first set of experiments included multiple blocks, with 25 trials each, when one odorant at one intensity was repeatedly presented. Each trial in the block included a 4-s stimulus presentation window. The inter-stimulus interval (between trials) was 60 s. Different odorants at different intensities were repeatedly presented in different blocks. A 15 min window, when no stimulus was presented, separated two consecutive blocks of trials. This window was included to reset any short-term memory that may have formed due to repeated presentation of the same stimulus (*Stopfer and Laurent, 1999*).

The second set of experiments involved two blocks of trials. The first block of trials included 30 trials, and the second block consisted of ten trials. In the first 25 trials of block 1, hexanol at 1% v/v was repeatedly presented. In the 26th trial, a puff of either isoamyl acetate 1% v/v or apple 1% v/v (a 'deviant' odorant) was presented. In trials 27–30, hexanol at 1% v/v was again presented. This was followed by a 15 min reset window when no stimulus was presented. In the second block of 10 trials

(trials 31–40), the deviant stimulus was repeatedly presented. All inter-stimulus intervals (between trials) were 60 s, and all odor presentations were four seconds long.

## Electrophysiology

Post-fifth instar adult locusts (*Schistocerca americana*) were reared in a crowded colony with a 12-hr–12-hr light-dark cycle. Both males and females were used for electrophysiological experiments. First, the locusts were immobilized with both antennae intact. Then the primary olfactory region of their brain, the antennal lobes, was exposed, desheathed, and perfused with room-temperature locust saline. Extracellular multiunit recordings of PNs were performed with a 16-channel, 4x4 silicon probe (NeuroNexus) that was superficially inserted in the antennal lobe (AL). Prior to each experiment, all probes were electroplated with gold to achieve impedances in the range of 200–300 kΩ. The recordings were acquired with a custom 16-channel amplifier (Biology Electronics Ship; Caltech, Pasadena, CA). The signals were amplified with a 10 k gain, bandpass filtered (0.3–6 kHz), and sampled at 15 kHz using a LabVIEW data acquisition system. A visual demonstration of this protocol is available online (*Saha et al., 2013a*).

## PN spike sorting

To obtain single-unit PN responses, spike sorting was performed offline using the four best recording channels and conservative statistical principles (*Pouzat et al., 2002*). Spikes belonging to single PNs were identified as described in earlier work (*Saha et al., 2013b*; *Saha et al., 2015*). The following criteria were used to identify single units: cluster separation >5 x noise standard deviations, number of cluster spikes within 20ms <6.5% of total spikes, and spike waveform variance <6.5 x noise standard deviations. In total, 161 PNs from 40 locusts were identified. Two datasets were collected. In the first dataset, responses of 80 PNs to hex and iaa were recorded, and in the second dataset, responses of 81 PNs to iaa and bza were monitored.

## Correlation analysis

The PN spikes were binned in 50ms non-overlapping time bins, and spike counts of different PNs were concatenated to obtain a population spike count vector. Pearson correlation coefficients between two PN ensemble spike count vectors were calculated using *Equation 1*.

$$C = \frac{cov\left(X^{trial\,i}, X^{trial\,j}\right)}{\sigma^{trial\,i} \cdot \sigma^{trial\,j}} \tag{1}$$

Here, $X^{trial\,i}$ and are time-averaged high-dimensional activity in *trials i* and *j*, respectively. $\sigma^{trial\,i}$ and $\sigma^{trial\,j}$ are the standard deviations of $X^{trial\,i}$ and $X^{trial\,j}$, respectively.

Each pixel/matrix element in the correlation plot shown in *Figure 5a* indicates the similarity between PN spike count vectors observed in the *i*th and *j*th trials.

## Tensor-based data decomposition

We first organized neural response data as a three-way array (Neuron ×Time × Trials; the stimulus information was also blended into the trial dimension) and then employed a direct three-way tensor decomposition approach (*Bro and PARAFAC, 1997*; *Bro and Kiers, 2003*). Here, the 3-D data cube was approximated using three loading matrices, A, B, and C, with elements $a_{if}$ (neuron dimension), $b_{jf}$ (time dimension), and $c_{kf}$ (trial dimension). $e_{ijk}$ was the residual element (see the equation below). The tri-linear model was found using alternating least squares.

$$x_{ijk} = \sum_{f=1}^{F} a_{if}b_{jf}c_{kf} + e_{ijk} \tag{2}$$

where $i, j, k$ denote the three different dimensions, and *F* indicates the total number of factors used for the analysis that was determined by the core consistency diagnostics (*Bro and PARAFAC, 1997*; *Bro and Kiers, 2003*). In our case, when $F = 3$, the core consistency was above 50 %, while it dropped to below 40% when $F = 4$. Therefore, we used three factors for our data decomposition.

## Trial-to-trial odor trajectory

For this analysis, we first reconstructed the dataset by computing the outer product of the loading matrices that were obtained by the tensor decomposition. The reconstructed 3-D tensor was then unfolded into a concatenated matrix (i.e. along the trial dimension). After unfolding, the ensemble PN responses were arranged as time series data of $n$ dimensions (where $n$ is the number of neurons) and $m$ steps (the number of 50 ms time bins × the number of trials). Note that only the PN activities during the 4-s stimulus presentation window in each trial were used for this analysis. The ensemble PN response vectors (in a given 50 ms time bin) were projected onto the three eigenvectors of the response covariance matrix that accounted for the most variance in the dataset, using principal component analysis. Finally, the low-dimensional points were connected in a temporal order to visualize neural response trajectories to different stimuli on a trial-to-trial basis. All trajectory plots shown in *Figure 4* were generated after smoothing with a three-point running average low-pass filter.

## Neural response similarity and dendrogram analysis

First, we calculated the summed spike counts during the 4 s odor presentation window for each PN. Then, the correlation similarity between two spike count profiles across PNs was calculated using *Equation 1*. Similarly, in this analysis, $x_i$ and $x_j$ represent a $n \times 1$ vector (n=80 for hex-2oct; n=81 for iaa-bza) for trial $i$ and $j$, respectively. The dendrogram was generated by hierarchical clustering of all stimulus identities, intensities, and individual trials based on the correlation distance. The dendrogram was created in such a way that the furthest pairwise distance between any two samples assigned to an individual cluster was minimized.

## Behavior experiments

Experiments were performed on post-fifth instar locusts of either sex that were starved for approximately 24 hr. All behavioral experiments occurred between 10 am and 2 pm. The protocols for the innate behavioral preference experiments and palp tracking used in this study were published in previous studies (*Saha et al., 2013b*; *Chandak and Raman, 2021*; *Nizampatnam et al., 2018*; *Saha et al., 2015*), where the locusts' thorax and legs were immobilized in custom casings that permitted the locusts' head, and most notably, their palps and antennae, to freely gesticulate. The odor delivery setup and stimulus sequences for the single odorants were similar to the ones described for the electrophysiological experiments. Any palp movement that occurred within 15 s of the odor stimulus onset was considered a POR. All data were hand-scored. Refer to *Figure 1—video 1* for a representative trial showing the POR response of a locust to hexanol (1% v/v) presentation.

## Acknowledgements

We thank members of the Raman Lab (Washington University in St. Louis) for feedback on the manuscript. We thank Pearl Olsen for insect care. This research was supported by NSF (1707221, 1724218, 2021795, 2319060) and ONR (N00014-19-1-2049, N00014-21-1-2343) grants to BR and an Imaging Sciences Fellowship to DL.

## Additional information

### Funding

| Funder | Grant reference number | Author |
|---|---|---|
| National Science Foundation | 1707221 | Baranidharan Raman |
| National Science Foundation | 1724218 | Baranidharan Raman |
| National Science Foundation | 2021795 | Baranidharan Raman |
| National Science Foundation | 2319060 | Baranidharan Raman |

| Funder | Grant reference number | Author |
|---|---|---|
| Office of Naval Research | N00014-19-1-2049 | Baranidharan Raman |
| Office of Naval Research | N00014-21-1-2343 | Baranidharan Raman |
| Washington University in St. Louis | Imaging Sciences Fellowship | Doris Ling |

The funders had no role in study design, data collection and interpretation, or the decision to submit the work for publication.

## Author contributions

Doris Ling, Investigation, Methodology, Writing – original draft, Writing – review and editing; Lijun Zhang, Formal analysis; Debajit Saha, Investigation, Methodology; Alex Bo-Yuan Chen, Investigation; Baranidharan Raman, Conceptualization, Formal analysis, Supervision, Funding acquisition, Writing – original draft, Project administration, Writing – review and editing

## Author ORCIDs

Debajit Saha ⬤ https://orcid.org/0000-0003-3136-9946
Alex Bo-Yuan Chen ⬤ https://orcid.org/0000-0003-3950-4460
Baranidharan Raman ⬤ https://orcid.org/0000-0002-7866-155X

Reviewer #1 (Public review): https://doi.org/10.7554/eLife.89330.3.sa1
Reviewer #3 (Public review): https://doi.org/10.7554/eLife.89330.3.sa2
Author response https://doi.org/10.7554/eLife.89330.3.sa3

## Additional files

### Supplementary files

MDAR checklist

### Data availability

Data and code made publicly available in figshare.

The following dataset was generated:

| Author(s) | Year | Dataset title | Dataset URL | Database and Identifier |
|---|---|---|---|---|
| Ling D, Zhang L, Saha D, Chen AB, Raman B | 2025 | Adaptation invariant concentration discrimination in an insect olfactory system | https://doi.org/10.6084/m9.figshare.28999229 | figshare, 10.6084/m9.figshare.28999229 |

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
