## [Editor Report · eLife Assessment]

This study addresses an **important** question in sensory neuroscience: how the olfactory system distinguishes decreases in stimulus intensity from decreases in neural responses due to adaptation. Based on a combination of electrophysiological and behavioral analyses, **solid** evidence establishes that neural coding changes differently between intensity reductions and adaptation, with intensity changes altering which neurons are activated while adaptation preserves the active ensemble but reduces response magnitude. Intriguingly, behavioral responses tend to increase as the neural responses decrease, suggesting that core features of the odor response persist through adaptation. While the experimental results are **convincing** overall, the conclusions will be strengthened by future work recording behavior and neural dynamics in the same animals.

---

## [Referee Report · Reviewer #1 (Public review)]

The authors use electrophysiological and behavioral measurements to examine how animals could reliably determine odor intensity/concentration across repeated experience. Because stimulus repetition leads to short-term adaptation evidenced by reduced overall firing rates in the antennal lobe and firing rates are otherwise concentration-dependent, there could be an ambiguity in sensory coding between reduced concentration or more recent experience. This would have a negative impact on the animal's ability to generate adaptive behavioral responses that depend odor intensities. The authors conclude that changes in concentration alter the constituent neurons contributing to the neural population response, whereas adaptation maintains the 'activated ensemble' but with scaled firing rates. This provides a neural coding account of the ability to distinguish odor concentrations even after extended experience. Additional analyses attempt to distinguish hypothesized circuit mechanisms for adaptation. A larger point that runs through the manuscript is that overall spiking activity has an inconsistent relationship with behavior and that the structure of population activity may be the more appropriate feature to consider.

To my knowledge, the dissociation of effects of odor concentration and adaptation on olfactory system population codes was not previously demonstrated. This is a significant contribution that improves on any simple model based on overall spiking activity. The primary result is most strikingly supported by visualization of a principal components analysis in Figure 4. Additional experiments and analysis complement and provide context for this finding regarding the relationship between neural population changes and behavior. There are some natural limitations on the interpretation of these data imposed by the methodology.

(1) Because individual recordings do not acquire a sufficient cell population to carry our population analyses, the cells must be combined into pseudopopulations for many analyses. This is common practice but it limits the ability to test the repeatability of findings across animals or populations. One potential additional solution would be to subsample the pseudopopulation, which would reveal the importance of individual sampled cells in the overall result. The utility of this additional testing is suggested by, for example, the benzaldehyde responses in supplementary figure 5, where two cells differentiate high and low concentration responses and would be expected to strongly impact correlation and classifier analyses.

(2) I do not think the analysis in Figure 2e can be strongly interpreted in terms of the vesicle depletion model. The hard diagonal bound on the lower part of each scatter plot indicates that features of the data/analysis necessarily exclude data in the lower left quadrant. I think this could be possibly explained by a floor effect wherein lower-response neurons cannot possibly express a large deltaResponse. To strengthen this case, one would need to devise a control analysis for the case where neural responses are simply all going as far down as they can go.

(3) Very minor, but it is confusing and not well-described how the error is computed in Figure 1f. One can imagine that the mean p(POR) is arrived at by averaging the binary values across locusts. Is this the case? If so, the same estimation of variance could be applied to Figures 1d and e

---

## [Referee Report · Reviewer #3 (Public review)]

Summary:

How does the brain distinguish stimulus intensity reduction from response reductions due to adaptation? Ling et al study whether and how the locust olfactory system encodes stimulus intensity and repetition differently. They show that these stimulus manipulations have distinguishable effects on population dynamics.

Strengths:

(1) Provides a potential strategy with which the brain can distinguish intensity decrease from adaptation. -- while both conditions reduce overall spike counts, intensity decrease can also changes which neurons are activated and adaptation only changes the response magnitude without changing the active ensemble.

(2) By interleaving a non-repeated odor, they show that these changes are odor-specific and not a non-specific effect.

(3) Describes how proboscis orientation response (POR) changes with stimulus repetition., Unlike the spike counts, POR increases in probability with stimulus. The data portray the variability across subjects in a clear way.

Weaknesses:

While POR and physiology can show a nice correlation when measured in different animals, additional insight would be gained from acquiring behavior and physiology simultaneously.

---

## [Author Response]

The following is the authors’ response to the original reviews.

**Reviewer #1 (Public Review):**
The authors use electrophysiological and behavioral measurements to examine how animals could reliably determine odor intensity/concentration across repeated experiences. Because stimulus repetition leads to short-term adaptation evidenced by reduced overall firing rates in the antennal lobe and firing rates are otherwise concentration-dependent, there could be an ambiguity in sensory coding between reduced concentration or more recent experience. This would have a negative impact on the animal's ability to generate adaptive behavioral responses that depend on odor intensities. The authors conclude that changes in concentration alter the constituent neurons contributing to the neural population response, whereas adaptation maintains the 'activated ensemble' but with scaled firing rates. This provides a neural coding account of the ability to distinguish odor concentrations even after extended experience. Additional analyses attempt to distinguish hypothesized circuit mechanisms for adaptation but are inconclusive. A larger point that runs through the manuscript is that overall spiking activity has an inconsistent relationship with behavior and that the structure of population activity may be the more appropriate feature to consider.To my knowledge, the dissociation of effects of odor concentration and adaptation on olfactory system population codes was not previously demonstrated. This is a significant contribution that improves on any simple model based on overall spiking activity. The primary result is most strikingly supported by visualization of a principal components analysis in Figure 4. However, there are some weaknesses in the data and analyses that limit confidence in the overall conclusions.

We thank the reviewer for evaluating our work and highlighting its strengths and deficiencies. We have revised the manuscript with expanded behavioral datasets and additional analyses that we believe convincingly support our conclusion.

(1) Behavioral work interpreted to demonstrate discrimination of different odor concentrations yields inconsistent results. Only two of the four odorants follow the pattern that is emphasized in the text (Figure 1F). Though it's a priori unlikely that animals are incapable of distinguishing odor concentrations at any stage in adaptation, the evidence presented is not sufficient to reach this conclusion.

We have expanded our dataset and now show that the behavioral response is significantly different for high and low concentration exposures of the same odorant. This was observed for all four odorants in our study (refer to Revised Fig. 1F).

(2) While conclusions center on concepts related to the combination of activated neurons or the "active ensemble", this specific level of description is not directly demonstrated in any part of the results. We see individual neural responses and dimensional reduction analyses, but we are unable to assess to what extent the activated ensemble is maintained across experience.

We have done several additional analyses (see provisional response). Notably, we have corroborated our dimensionality reduction and correlation analysis results with a quantitative classification analysis that convincingly demonstrates that odor identity and intensity of the odorant can be decoded from the ensemble neural activity, and this could be achieved in an adaptation-invariant fashion (refer to Revised Supplementary Fig. 4).

(3) There is little information about the variance or statistical strength of results described at the population level. While the PCA presents a compelling picture, the central point that concentration changes and adaptation alter population responses across separable dimensions is not demonstrated quantitatively. The correlation analysis that might partially address this question is presented to be visually interpreted with no additional testing.

We have included a plot that compares the odor-evoked responses across all neurons (mean ± variance) at both intensity levels for each odorant (Revised Supplementary Fig. 5). This plot clearly shows how the ensemble neural activity profile varies with odor intensity and how these response patterns are robustly maintained across trials.

(4) Results are often presented separately for each odor stimulus or for separate datasets including two odor stimuli. An effort should be made to characterize patterns of results across all odor stimuli and their statistical reliability. This concern arises throughout all data presentations.

We had to incorporate a 15-minute window between presentations of odorants to reset adaptation. Due to this, we were unable to extracellularly record from all four odorants at two intensities from a single experiment (~ 3.5 hours of recording for just 2 odorants at two intensities with one odorant at higher intensity repeated at the end; Fig. 2a). Therefore, we recorded two datasets. Each dataset captured the responses of ~80 PNs to two odorants at two intensities, one odorant at the higher concentration repeated at the end of the experiment to show repeatability of changes due to adaptation.

(5) The relevance of the inconclusive analysis of inferred adaptation mechanisms in Figure 2d-f and the single experiment including a complex mixture in Figure 7 to the motivating questions for this study are unclear.

Figure 2d-f has been revised. While we agree that the adaptation mechanisms are not fully clear, there is a trend that the most active PNs are the neurons that change the most across trials. This change and the response in the first trial are negatively correlated, indicating that vesicle depletion could be an important contributor to the observed results. However, neurons that adapt strongly at higher intensities are not the ones that adapt at lower intensities. This complicates the understanding of how neural responses vary with intensities and the adaptation that happens due to repetition. This has been highlighted in the revised manuscript.

Regarding Figure 7, we wanted to examine the odor-specificity of the changes that happen due to repeated encounters of an odorant. Specifically, wondered if the neural response reduction and behavioral enhancements were a global, non-specific state change in the olfactory system brought about by the repetition of any odorant, or are the observed neural and behavioral response changes odor-specific.

(6) Throughout the description of the results, typical standards for statistical reporting (sample size, error bars, etc.) are not followed. This prevents readers from assessing effect sizes and undermines the ability to assign a confidence to any particular conclusion.

We have revised the manuscript to fix these issues and included sample size and error bars in our plots.

**Reviewer #2 (Public Review):**
Summary:The authors' main goal was to evaluate how both behavioral responses to odor, and their early sensory representations are modified by repeated exposure to odor, asking whether the process of adaptation is equivalent to reducing the concentration of an odor. They open with behavioral experiments that actually establish that repeated odor presentation increases the likelihood of evoking a behavioral response in their experimental subjects - locusts. They then examine neural activity patterns at the second layer of the olfactory circuit. At the population level, repeated odor exposure reduces total spike counts, but at the level of individual cells there seems to be no consistent guiding principle that describes the adaptation-related changes, and therefore no single mechanism could be identified.Both population vector analysis and pattern correlation analysis indicate that odor intensity information is preserved through the adaptation process. They make the closely related point that responses to an odor in the adapted state are distinct from responses to lower concentration of the same odor. These analyses are appropriate, but the point could be strengthened by explicitly using some type of classification analysis to quantify the adaptation effects. e.g. a confusion matrix might show if there is a gradual shift in odor representations, or whether there are trials where representations change abruptly.Strengths:One strength is that the work has both behavioral read-out of odor perception and electrophysiological characterization of the sensory inputs and how both change over repeated stimulus presentations. It is particularly interesting that behavioral responses increase while neuronal responses generally decrease. Although the behavioral effect could occur fully downstream of the sensory responses the authors measure, at least those sensory responses retain the core features needed to drive behavior despite being highly adapted.Weaknesses:Ultimately no clear conceptual framework arises to understand how PN responses change during adaptation. Neither the mechanism (vesicle depletion versus changes in lateral inhibition) nor even a qualitative description of those changes. Perhaps this is because much of the analysis is focused on the entire population response, while perhaps different mechanisms operate on different cells making it difficult to understand things at the single PN level.From the x-axis scale in Fig 2e,f it appeared to me that they do not observe many strong PN responses to these stimuli, everything being < 10 spikes/sec. So perhaps a clearer effect would be observed if they managed to find the stronger responding PNs than captured in this dataset.

We thank the reviewer for his/her evaluation of our work. Indeed, our work does not clarify the mechanism that underlies the adaptation over trials, and how this mechanism accounts for adaptation that is observed at two different intensities of the same odorant. However, as we highlight in the revised manuscript, there is some evidence for the vesicle depletion hypothesis. For the plots shown in Fig. 2, the firing rates were calculated after averaging across time bins and trials. Hence, the lower firing rates. The peak firing rates of the most active neurons are ~100 Hz. So, we are certain that we are collecting responses from a representative ensemble of neurons in this circuit.

**Reviewer #3 (Public Review):**
Summary:How does the brain distinguish stimulus intensity reduction from response reductions due to adaptation? Ling et al study whether and how the locust olfactory system encodes stimulus intensity and repetition differently. They show that these stimulus manipulations have distinguishable effects on population dynamics.Strengths:(1) Provides a potential strategy with which the brain can distinguish intensity decrease from adaptation. -- while both conditions reduce overall spike counts, intensity decrease can also changes which neurons are activated and adaptation only changes the response magnitude without changing the active ensemble.(2) By interleaving a non-repeated odor, they show that these changes are odor-specific and not a non-specific effect.(3) Describes how proboscis orientation response (POR) changes with stimulus repetition., Unlike the spike counts, POR increases in probability with stimulus. The data portray the variability across subjects in a clear way.

We thank the reviewer for the summary and for highlighting the strengths of our work.

Weaknesses:(1) Behaviora. While the "learning curve" of the POR is nicely described, the behavior itself receives very little description. What are the kinematics of the movement, and do these vary with repetition? Is the POR all-or-nothing or does it vary trial to trial?

The behavioral responses were monitored in unconditioned/untrained locusts. Hence, these are innate responses to the odorants. These innate responses are usually brief and occur after the onset of the stimulus. However, there is variability across locusts and trials (refer Revised Supplementary Fig. 1). When the same odorant is conditioned with food reward, the POR responses become more stereotyped and occur rapidly within a few hundred milliseconds.

**Author response image 1. sa3fig1:** POR response dynamics in a conditioned locust. The palps were painted in this case (left panel), and the distance between the palps was tracked as a function of time (right panel).

b. What are the reaction times? This can constrain what time window is relevant in the neural responses. E.g., if the reaction time is 500 ms, then only the first 500 ms of the ensemble response deserves close scrutiny. Later spikes cannot contribute.

This is an interesting point. We had done this analysis for conditioned POR responses. For innate POR, as we noted earlier, there is variability across locusts. Many responses occur rapidly after odor onset (<1 s), while some responses do occur later during odor presentation and in some cases after odor termination. It is important to note that these dynamical aspects of the POR response, while super interesting, should occur at a much faster time scale compared to the adaptation that we are reporting across trials or repeated encounters of an odorant.

c. The behavioral methods are lacking some key information. While references are given to previous work, the reader should not be obligated to look at other papers to answer basic questions: how was the response measured? Video tracking? Hand scored?

We agree and apologize for the oversight. We have revised the methods and added a video to show the POR responses. Videos were hand-scored.

d. Can we be sure that this is an odor response? Although airflow out of the olfactometer is ongoing throughout the experiment, opening and closing valves usually creates pressure jumps that are likely to activate mechanosensors in the antennae.

Interesting. We have added a new Supplementary Fig. 2 that shows that the POR to even presentations of paraffin oil (solvent; control) is negligible. This should confirm that the POR is a behavioral response to the odorant.

Furthermore, all other potential confounds identified by the reviewer are present for every odorant and every concentration presented. However, the POR varies in an odor-identity and intensity-specific manner.

e. What is the baseline rate of PORs in the absence of stimuli?

Almost zero.

f. What can you say about the purpose of the POR? I lack an intuition for why a fly would wiggle the maxillary palps. This is a question that is probably impossible to answer definitively, but even a speculative explanation would help the reader better understand.

The locusts use these finger-like maxillary palps to grab a grass blade while eating. Hence, we believe that this might be a preparatory response to feeding. We have noted that the PORs are elicited more by food-related odorants. Hence, we think it is a measure of odor appetitiveness. This has been added to the manuscript.

(2) Physiologya. Does stimulus repetition affect "spontaneous" activity (i.e., firing in the interstimulus interval?) To study this question, in Figures 2b and c, it would be valuable to display more of the prestimulus period, and a quantification of the stability or lability of the inter-stimulus activity.

Done. Yes, the spontaneous activity does appear to change in an odor-specific manner. We have done some detailed analysis of the same in this preprint:

Ling D, Moss EH, Smith CL, Kroeger R, Reimer J, Raman B, Arenkiel BR. Conserved neural dynamics and computations across species in olfaction. bioRxiv [Preprint]. 2023 Apr 24:2023.04.24.538157. doi: 10.1101/2023.04.24.538157. PMID: 37162844; PMCID: PMC10168254

b. When does the response change stabilize? While the authors compare repetition 1 to repetition 25, from the rasters it appears that the changes have largely stabilized after the 3rd or 4th repetition. In Figure 5, there is a clear difference between repetition 1-3 or so and the rest. Are successive repetitions more similar than more temporally-separated repetitions (e.g., is rep 13 more similar to 14 than to 17?). I was not able to judge this based on the dendrograms of Figure 5. If the responses do stabilize at it appears, it would be more informative to focus on the dynamics of the first few repetitions.

The reviewer makes an astute observation. Yes, the changes in firing rates are larger in the first three trials (Fig. 3c). The ensemble activity patterns, though, are relatively stable across all trials as indicated by the PCA plots and classification analysis results.

**Author response image 2. sa3fig2:** Correlation as a function of trial number. All correlations were made with respect to the odor-evoked responses in the last odor trial of hex(H) and bza(H).

c. How do temporal dynamics change? Locust PNs have richly varied temporal dynamics, but how these may be affected is not clear. The across-population average is poorly suited to capture this feature of the activity. For example, the PNs often have an early transient response, and these appear to be timed differently across the population. These structures will be obscured in a cross population average. Looking at the rasters, it looks like the initial transient changes its timing (e.g., PN40 responses move earlier; PN33 responses move later.). Quantification of latency to first spike after stimulus may make a useful measure of the dynamics.

As noted earlier, to keep our story simple in this manuscript, we have only focused on the variations across trials (i.e., much slower response dynamics). We did this as we are not recording neural and behavioral responses from the same locust. We plan to do this and directly compare the neural and behavioral dynamics in the same locust.

d.How legitimate is the link between POR and physiology? While their changes can show a nice correlation, the fact the data were taken from separate animals makes them less compelling than they would be otherwise. How feasible is it to capture POR and physiology in the same prep?This would be most helpful, but I suspect may be too technically challenging to be within scope.

The antennal lobe activity in the input about the volatile chemicals encountered by the locust. The POR is a behavioral output. Hence, we believe that examining the correlation between the olfactory system's input and output is a valid approach. However, we have only compared the mean trends in neural and behavioral datasets, and dynamics on a much slower timescale. We are currently developing the capability to record neural responses in behaving animals. This turned out to be a bit more challenging than we had envisioned. We plan to do fine-grained comparisons of the neural and behavioral dynamics, recommended by this reviewer, in those preparations.

Further, we will also be able to examine whether the variability in behavioral responses could be predicted from neural activity changes in that prep.